

Annual changes in plant functional types and their responses to climate
change on the Northern Tibetan Plateau
Lan Cuo[1], Yongxin Zhang[2], Shilong Piao[3] Yanhong Gao[4]
[1]Center for Excellence in Tibetan Plateau Earth Sciences, Key Laboratory of Tibetan
Environment Changes and Land Surface Processes, Institute of Tibetan Plateau Research,
Chinese Academy of Sciences, Beijing, China,
[2]Research Applications Laboratory, National Center for Atmospheric Research, Boulder,
Colorado, USA
[3]Center for Excellence in Tibetan Plateau Earth Sciences, Key Laboratory of Tibetan
Environment Changes and Land Surface Processes, Institute of Tibetan Plateau Research,
Chinese Academy of Sciences, Beijing, China
[4] Key Laboratory of Land Surface Process and Climate Change in Cold and Arid Regions,
Cold and Arid Regions Environmental and Engineering Research Institute, Chinese
Academy of Sciences, Lanzhou, China
Corresponding author:
Lan Cuo
Phone: 086-010-84097091
Fax: 086-010-84097079
Email: lancuo@itpcas.ac.cn





**Abstract** Changes in plant functional types (PFTs) have important implications for both
climate and water resources. Still, little is known about whether and how PFTs have
changed over the past decades on the Northern Tibetan Plateau (NTP) where several of
the top largest rivers in the world are originated. Also, the relative importance of
atmospheric conditions versus soil physical conditions in affecting PFTs is unknown on
the NTP. In this study, we used the improved Lund-Potsdam-Jena Dynamic Global
Vegetation Model to investigate PFT changes through examining the changes in foliar
projective coverages (FPCs) during 1957-2009 and their responses to changes in root
zone soil temperature, soil moisture, air temperature, precipitation and $CO_2$
concentrations. The results show spatially heterogeneous changes in FPCs across the
NTP during 1957-2009, with 34% (13%) of the region showing increasing (decreasing)
trends. Dominant drivers responsible for the observed FPC changes vary with regions and
vegetation types, but overall, precipitation is the major factor in determining FPC
changes on the NTP with positive impacts. Soil temperature increase exhibits small but
negative impacts on FPCs. Different responses of individual FPCs to regionally varying
climate change result in spatially heterogeneous patterns of vegetation changes on the
NTP. The implication of the study is that fresh water resources in one of the world's
largest and most important headwater basins and the onset and intensity of Asian
monsoon circulations could be affected because of the changes in FPCs on the NTP.
**Keywords**: Plant functional types, foliar projective coverage, dynamic vegetation
modeling, climate change, northern Tibetan Plateau, desertification



## 1. Introduction

Vegetation dynamics can directly affect water, energy and carbon balances in the coupled

land-atmosphere system by responding and providing feedbacks to climate change

(Bonan et al., 1992; Bonan et al., 2003; Rogers et al., 2013; Ahlstrom et al., 2015;

Mengis et al., 2015; Paschalis et al., 2015; Peterman et al., 2015; Sitch et al., 2015). In

recent years, dynamic global vegetation models (DGVMs) coupled with atmospheric

processes have become valuable tools for examining and understanding the interactive

dynamics in carbon, water, and energy exchanges between biosphere and atmosphere.

The representation of dynamic vegetation has also become a key component in the earth

system models since the last decade (Levis et al., 2004; Sato et al., 2007; Hopcroft et al.,

2015). There are many widely used DGVMs that include TRIFFID (Cox, 2001), LPJ

(Sitch et al., 2003), BIOME-BGC (Tatarinov and Cienciala, 2006), CENTURY

(Smithwick et al., 2009), and OCHIDEE (Ciais et al, 2008), just to name a few. Most of

these DGVMs employ the so-called climate envelop approach to control the

redistribution of plant, whereas TRIFFID uses the Lotka-Volterra representation of

competitive ecological processes for plant redistribution (Fisher et al., 2015).

One common way to describe vegetation in many DGVMs is the adoption of the term

plant functional type (PFT) for classifying plants into discrete groups according to their

ecological, physiological and phylogenetic traits (Cox, 2001; Sitch et al., 2003).  In many

of the state-of-the-art hydrological and land surface models, PFT is both an input for

driving the land-atmosphere processes (e.g., Liang et al., 1994; Liang et al., 1996;

Wigmosta et al., 1994) and an output of dynamic vegetation simulations (e.g., Sitch et al,



2003). When climate changes, PFTs may migrate or retreat depending on bioclimatic
limits and availability of water and light (Pearson et al., 2013). For example, Jiang et al.
(2012) examined the Lund-Potsdam-Jena Dynamic Global Vegetation Model (LPJ-
DGVM) simulations and showed that temperate trees were more sensitive to climate
change than boreal trees and perennial C3 grasses, suggesting that anomalous warming in
the northern high latitudes could change the composition of PFTs and cause the
northward migration of temperate trees. Changes in the composition of PFTs due to
climate change could also modify the foliar projective coverage (FPC, the proportion of
ground area that is covered by leaves), an important quantity in determining water,
energy, and C exchange (Weiss et al, 2014; Meng et al., 2015). Given the fact that
evapotranspiration and photosynthesis are closely related to the foliar properties (Swank
and Douglass, 1974; Huber and Iroume, 2001; Zhang et al., 2015), some dynamic
vegetation models use FPC to represent PFT (e.g., Sitch et al., 2003).

Due to its massive size and high altitude, the Tibetan Plateau (TP) exerts strong influence
on regional and global climate through mechanical and thermal-dynamic forcing (Yanai
et al., 1992; Wang et al., 2008). The TP is characterized by complex terrain,
heterogeneous land surfaces, spatially varying energy and water patterns, diverse
ecosystems and bioclimatic zones (Yeh and Gao, 1979; CAS, 2007). The surface
conditions of the TP, such as snow cover, soil moisture and vegetation all affect the
strength and evolution of the East Asian and South Asian monsoons (Reiter and Gao,
1982; Ye and We, 1998; Zhang et al., 2004; Qiu, 2008).  It is also the headwater region
of the major rivers in Asia (Cuo et al., 2014). In particular, the northern TP (NTP; 30-



40 °N, 90-105°E) where the Yellow, Yangtze and Mekong Rivers originate, is crucially
important in providing water and other ecosystem services to the plateau itself and the
downstream regions hosting billions of population. Changes in the composition of
different PFTs, and consequently FPCs, could substantially affect surface
evapotranspiration, soil water storage and streamflow (Cuo et al., 2009; Cuo et al., 2013a;
Weiss et al., 2014; Cuo, 2015; Dahlin et al., 2015), and the partition of net radiation into
the sensible and latent heat fluxes, consequently affecting the onset and intensity of south
and east Asian monsoon circulations (Wu et al., 2007; Cui et al., 2015). Although there
are some studies that connect NDVI (Normalized Difference Vegetation Index) and NPP
(Net Primary Production) to precipitation, air temperature, and $CO_2$ concentrations on the
TP (Zhong et al., 2010; Chen et al., 2012; Piao et al., 2012), very few studies have
examined PFT changes and their relationships with climate for the region (Wang X.,

105  2011).


Besides precipitation, air temperature and $CO_2$ change impacts on the plants, changes in
soil temperature and soil moisture could also affect heterotrophic respiration (litter
decomposition, soil carbon release, etc.) and vegetation root development. Jin et al. (2013)
found that spatial patterns and temporal trends of phenology were parallel with the
corresponding soil physical conditions over the TP, and that 1°C increase in soil
temperature could advance the start of the growing season by 4.6 – 9.9 days. On the TP
where a vast area of seasonally frozen (SFS) and permafrost (PFS) soil exists (Cheng and
Jin, 2013), global warming induced frozen soil degradation (Cuo et al., 2015) could
potentially affect litter decomposition and plant phenology (Jin et al., 2013).




To date, little is still known about the changes in PFTs or FPC on the NTP in recent
decades, much less the mechanisms behind these changes, largely due to the lack of long-
term observation data and appropriate research tools. This knowledge gap greatly limits
our understanding of TP's vegetation dynamics in response to climate change and the
associated implications in the regional and global water, energy and carbon cycles. This
study aims to fill this knowledge gap by investigating the changes in PFTs on the NTP in
1957-2009 and the underlying mechanisms using a dynamic vegetation model, the LPJ-
DGVM model. Important atmospheric and soil variables that could significantly affect
PFT changes, including precipitation, air temperature, $CO_2$ concentration, soil
temperature and soil moisture, are examined and their importance is compared using a
dynamic vegetation model.

**2. Methods and data**
2.1 Study area
The NTP lies between 1400 and 6100 m above sea level, with an average elevation of
around 3900 m (Fig. 1). Five large mountains, the Hengduan in the southeast, the
Tanggula in the southwest, the Kunlun in the center, the Arjin in the northwest, and the
Qilian in the north are located on the NTP. Vegetation on the NTP changes from forest in
the southeast to grassland and desert in the northwest, with major vegetation types
including temperate evergreen needleleaf forest, summergreen needleleaf and broadleaf
forest, temperate shrub/grassland, alpine meadow, alpine steppe, sparsely vegetated bare
land and desert. Annual precipitation ranges from 1000 mm in the southeast to less than



100 mm in the northwest. Annual air temperature is high in the low elevation (about
15 °C) and low in the high elevation (about -10 °C). Details of the spatial patterns of the
climate elements and their changes on the NTP over the past five decades can be found in
Cuo et al. (2013b).

2.2 The LPJ model and its parameterizations for the NTP
We used the LPJ-DGVM model (Sitch et al., 2003; Gerten et al., 2004; LPJ hereafter) to
simulate vegetation dynamics, C cycle and biogeophysical properties. Vegetation
dynamics in LPJ are driven by the processes of competition for water, light and nutrients
among plant functional types, with different rates of plant carbon assimilation and
allocation, reproduction, and survival. LPJ can simulate photosynthesis, transpiration, soil
organic matter and litter dynamics and fire disturbance at daily time step, and resource
competition, tissue turnover, population dynamics at annual time step. Plant
establishment is determined by bioclimatic limits. Probability of plant mortality is
controlled by the interactions among light competition, low growth efficiency, a negative
carbon balance, heat stress and bioclimatic limits. LPJ couples fast hydrological and
physiological processes with slower ecosystem processes using daily, monthly, and
yearly time scales (Bonan et al., 2003), and has been successfully applied in the
simulation of global and regional vegetation dynamics and large scale PFT distributions
(Smith et al., 2001; Sitch et al., 2003; Gerten et al., 2004; Sitch et al., 2005; Sitch et al.,
2008; Murray, 2014; Steinkamp and Hickler, 2015).





Six PFTs, temperate needleleaf evergreen trees, temperate broadleaf evergreen trees,
temperate broadleaf summergreen trees, perennial alpine meadow grasses, perennial
alpine steppe grasses, and perennial temperate summergreen scrub/grassland are
compiled and used in the model to represent the major vegetation types on the NTP,
based on physiognomic (tree or herbaceous), bioclimatic (temperate, boreal or alpine),
phenological (evergreen or summergreen), and photosynthetic (C3 or C4) properties of
the plants. The vegetation state of each of the 0.25° × 0.25° grid cells in LPJ is a mixture
of PFTs that can be distinguished by their FPCs. FPC of an individual PFT, ranging from
0 (zero coverage) to 100 (full coverage), is a function of crown area (for trees only),
individual density and LAI, and is calculated by the Lambert-Beer law (Sitch et al., 2003).
The total FPC of a given space is the sum of the FPCs of all PFTs in that space.

On the NTP, vegetation root system is concentrated in the top 0.4 m depth where soil
undergoes seasonal freezing and thawing cycles. The accuracy of heat and moisture
content representation in the top 0.4 m soil is therefore vital for modeling vegetation
dynamics and C cycle in this region. In this work, LPJ is configured with two soil layers,
0-0.4 m (top layer) and 0.4-1.0 m (bottom layer) below surface, for better accounting for
water and energy states of the top soil layer under repeated freezing and thawing cycles
on the NTP, while at the same time maintaining its computational efficiency for large
scale simulations. Daily temperature of the top soil layer is calculated by linearly
regressing it with daily air temperature. The linear relationship is obtained from five
stations (stars in Fig. 1) where both soil temperature and air temperature observations are
available. These five stations are located in the different parts of the NTP and represent



various land cover types (temperate shrub/grassland, alpine meadow, alpine steppe and
desert) and soil conditions (SFS and PFS). Depending on the stations, the observation
periods are different. Both monthly and annual soil temperature at stations with
observation periods greater than 2 years are chosen for the validation of simulated soil
temperature. The linear regression equations are developed separately for normal (regular
soil moisture) and desert (dry) soils. For normal soil, daily soil (0-0.4 m depth) and air
temperature are obtained from Mengyuan (1983-2009, SFS), Maduo (1980-2009, SFS)
and Wudaoliang (2005-2006, PFS)  (Eq. 1). For desert dry soils where monthly soil
moisture is usually around 0.1 $m^3/m^3$, daily soil and air temperature are obtained from the
Mangai (1988-2009) and Lenghu (1980-2009) stations (Eq. 2). Note that desert dry soil
temperature can change quickly due to the lower thermal capacity of dry air (1000 J $K^{-1}$
$kg^{-1}$) than that of water (4188 J $K^{-1}$ $kg^{-1}$), and the slope for desert dry soil is larger than
that for normal soil. Eqs. (1) and (2) are expressed as:
$ST1 = 0.8753 \times AT + 3.1623, \ \theta > 0.1; \quad R^2 = 0.94$        (1)
$ST1 = 1.0873 \times AT + 3.9063, \ \theta \leq 0.1; \quad R^2 = 0.97$        (2)
where ST1 is daily soil temperature (°C) in 0-0.4 m depth, AT is daily air temperature
(°C), and θ is total soil moisture ($m^3/m^3$). $R^2$ is coefficient of determination.

Soil temperature in 0.4–1.0 m depth is assumed to be a lagged exponential function of the
top layer soil temperature. The equations are as follows:
$ST2 = ST1 + (Ta - ST1) \times e^{-Ts}$        (3)
$Ta = a + b \times \left( N_d - 1 - Ts \times \frac{365}{2\pi} \right)$        (4)



$$Ts = \frac{D_2 \times \frac{3}{4}}{\sqrt{Q_d \times 86400 \times 365/\pi}}$$ (5)
$$Q_d = \frac{K}{C_s}$$ (6)
where $K$ and $C_s$ are heat conductivity (W m$^{-1}$ K$^{-1}$) and volumetric heat capacity (J m$^{-3}$ K$^{-}$
$^1$), respectively, that are calculated based on the soil mineral and organic content and
moisture conditions and are updated at a daily time step; $Q_d$ is heat diffusivity (m$^2$ s$^{-1}$); $D_2$
is the depth of the second layer; $N_d$ is the number of days in a year; $a$ and $b$ are the linear
regression coefficients of daily air temperature and the numbers of days in a month,
respectively, and are updated at a monthly time step. Eq. (4) calculates the lag of the
thermal change in the second layer soil temperature. The equations employed for the
second layer soil temperature are the modified version of the originals used in LPJ.

Total soil moisture in the top soil layer is obtained from the balance between precipitation
input, soil evapotranspiration and percolation. Ice and liquid content is calculated based
on soil temperature. If soil temperature is below 0 °C, soil liquid content is calculated by
using freezing point depression equation. Ice content is the difference between total soil
moisture and liquid water content. When soil temperature is greater than 0 °C, soil
moisture is liquid and ice content is zero. The equations are:
$$\theta_l = \varphi \times \left(\frac{-L_f \times ST1}{273.16 \times g \times b_p}\right)^{-\frac{2.0}{n-3}} \qquad ST1 < 0$$ (7)
$$\theta_i = \theta - \theta_l \qquad ST1 < 0$$ (8)
where $\theta$ is total soil moisture and subscripts $l$ and $i$ represent liquid and ice, respectively;
$\varphi$ is soil porosity; $L_f$ is latent heat of fusion (3.337×10$^5$ J kg$^{-1}$); $g$ is gravitational
acceleration (9.81 m s$^{-2}$); $b_p$ is the bubbling pressure (m); and $n$ is the exponent in



Campbell's equation for hydraulic conductivity. The second layer soil moisture is
calculated using the similar equations, and it is the aggregation of liquid and ice content,
runoff, percolated moisture from the top layer and to the baseflow. Runoff is generated
when liquid soil content is greater than porosity, and percolation is generated when liquid
soil content is greater than soil water holding capacity. Runoff and baseflow are produced
in both soil layers and are removed from soil moisture. Soil moisture observations are
rare on the NTP. Only 1-year observations at 4 permafrost sites are available during the
study period (see Fig. 1) and they are used for soil moisture validation.

The implementation of the aforementioned processes in the LPJ model requires seven
additional soil parameters for each of the two soil layers: Campbell's exponent $n$,
bubbling pressure $b_p$, bulk densities for organic matter and soil mineral, particle densities
for organic matter and soil mineral, and quartz content. Soil porosity $\varphi$ is calculated from
soil bulk density and soil particle density. These parameters are often used in surface
hydrological models for calculating soil hydrological properties (e.g. Liang et al., 1994;
1996; Wigmosta et al., 2004), and are provided for various soil texture types in the LPJ
model. These additional parameters together with the original parameters for soil texture,
soil percolation rates and water holding capacity constitute the new soil parameter sets.
These modifications eliminate the use of fixed heat diffusivity at 0%, 15% and 100%
water content in the original model version, instead here the diffusivity varies with
thermal conductivity and capacity as shown in Eq. (6).

2.3 Forcing data and observations



Forcing data used in the LPJ model include monthly air temperature, precipitation, wet
days, cloud cover amount and annual $CO_2$ concentrations.  Monthly air temperature,
precipitation and wet days, all at 0.25° × 0.25° resolution, were from Cuo et al. (2013b).
Cloud cover data came from the Climate Research Unit of the University of East Anglia
(Mitchell and Jones, 2005) at 0.5° × 0.5° resolution and were regridded to 0.25° × 0.25°
resolution assuming uniform distribution of cloud cover within each 0.5° × 0.5° grid cell.
Annual $CO_2$ concentrations were obtained from the Mauna Loa Observatory operated by
NOAA (National Oceanic and Atmospheric Administration). Missing $CO_2$ observations
for 1957 - 1958 were filled in by extrapolating the regression between annual $CO_2$
concentrations and the corresponding years. Soil texture data were from the Harmonized
World Soil Data v1.0 (FAO, 2008). Elevation data were from the Shuttle Radar
Topography Mission (SRTM) and were interpolated to the 0.25° × 0.25° grids.

2.4 Analysis methods
To spin up, the LPJ model was run iteratively for 1000 years using the 1957-1986 climate
data and starting from bare ground, a common practice among LPJ users. The purpose of
this long run is to establish ecosystem equilibrium equivalent to the 1957 conditions. Like
earlier studies (e.g., Sitch et al., 2003), we assume that after the 1000-year spinup,
vegetation dynamics, carbon pools, soil thermal and water conditions reach the needed
equilibrium.

Given the importance of the top 0.4 m soils for vegetation root system on the NTP, we
validate model simulated soil temperature and moisture in this layer against available



observations. Deep layer soil temperature and moisture are also evaluated but will not be
shown. Mean, correlation coefficient (R) and root mean square error (RMSE) of monthly
and annual mean soil temperature, as well as monthly soil moisture are examined. FPCs
are used to represent vegetation states and PFTs. The spatial patterns of the simulated
FPCs of dominant PFTs are compared with those of the survey maps compiled by CAS
(2007) and Zheng et al. (2008). Parameters representing the physiological, phenological
and bioclimatic attributes of the six PFTs are adjusted accordingly to obtain a reasonable
match between the simulated pattern and the survey maps. Soil parameters used are from
Cuo et al. (2013a) and model default settings. The calibrated PFT parameters are listed in
Table 1.

Following model evaluation, we examine the changes in total FPCs and FPCs of
individual PFTs during 1957-2009 in response to climate change. Climate change is
represented by changes in air and 40-cm-deep soil temperatures, 40-cm-deep soil
moisture, precipitation and atmospheric $CO_2$ concentration. The Mann-Kendall trend
analysis is employed to investigate the FPC trends. Also, the differences between
historical simulation and climate trends removed simulation are examined to identify the
changes in FPCs during the past five decades.

To investigate the sensitivity of FPCs in each grid cell to changes in soil temperature and
moisture, air temperature, precipitation and $CO_2$, six scenarios (S1-S6) are designed
(Table 2). In the baseline scenario (S6), the trends in air temperature, precipitation, wet
day and $CO_2$ are removed. Soil temperature and moisture respond to atmospheric



forcings and they are assumed to have no trends when the trends of atmospheric forcings
are removed. The only difference between S1-5 and S6 scenarios is the introduced
change in one variable while keeping the other variables unchanged. Cloud cover remains
the same for all scenarios. For precipitation, only the amount but not the frequency is
changed. These scenarios bear similarity to what has been identified over the NTP in
recent decades in general in that S1 plus S2, S3, S4 and S5 represent regional frozen soil
degradation, warming, wetting and elevated $CO_2$ trends, respectively, although the rates
of changes and spatial patterns differ (Cuo et al., 2013b, 2015). Uniform perturbations are
introduced to provide the benchmark for the climate sensitivity comparison across the
region and to derive sensitivity spatial pattern. It is expected that the comparisons
between the paired S1-S6, S2-S6, S3–S6, S4-S6 and S5-S6 scenarios would reveal the
responses of FPC to the changes in soil temperature, soil moisture, air temperature,
precipitation and $CO_2$, respectively.

Using S1-6 scenarios, we examine elasticity (E), a quantity that measures how responsive
a variable is to a changing condition, in order to quantify the degree of the FPC
sensitivity to climate change. Elasticity is calculated as the median of the ratios of
percentage changes in annual FPC to the percentage changes in an annual climate
variable. Positive (negative) E indicates that FPC increases (decreases) with changing
climate variable. Larger E corresponds to higher sensitivity, and when E is zero FPC is
not responding to climate change. In the following, we will use $E_{ST1}$, $E_{sm1}$, $E_{AT}$, $E_{PRCP}$ and
$E_{CO2}$ to denote the sensitivity of FPCs to the changes in the top layer soil temperature



(ST1) and soil moisture (SM1), air temperature (AT), precipitation (PRCP), and $CO_2$,
respectively.

**3. Results**
3.1 Evaluations of simulated soil temperature, moisture and FPC
Figure 2 shows the simulated and observed monthly soil temperature in the top 0.4 m
depth at Wudaoliang (PFS), Maduo (SFS), Mengyuan (SFS), Mangai (SFS, dry desert
soil) and Lenghu (SFS, dry desert soil). The observations at these sites are also used to
derive the linear regression equations that are then applied over the entire domain. At
Maduo and Mengyuan, the simulated soil temperature matches the observed rather well
in both magnitude and seasonal cycles. At Wudaoliang, the highest station, the simulated
magnitude of the seasonal cycle in soil temperature is larger than the observed while the
opposite is true at Mangai and Lenghu, two dry desert soil stations. Correlation between
the simulations and observations is high (R ≥ 0.96) across these five stations and RMSE
ranges from 1.40 °C at Maduo to 3.07 ºC at Lenghu (Table 3).

As an independent check, we also compare the simulated and observed monthly soil
temperature of the top soil layer at five other stations whose observations are not used in
deriving the linear regression equations (Fig. 3). These five stations are Xidatan (PFS),
Xinghai (SFS), Zaduo (SFS), Qilian (SFS), and Golmud (SFS, dry desert soil). All five
stations except Xidatan show satisfactory simulations in both magnitude and seasonal
cycles when compared to the observations although Qilian displays underestimation in
the peaks while Golmud exhibits slight overestimation in the first half of the years. At



Xidatan, a PFS site, the simulated seasonal cycle is larger than the observed, similar to
Wudaoling as discussed before. It appears that the derived linear regression relationships
may contain some deficiencies at the PFS sites due to the limited observations in the PFS
region. RMSE ranges from 1.32 °C at Zaduo to 3.03 ºC at Golmud and 3.21 ºC at Xidatan
(Table 3). Correlation between the simulated and observed monthly soil temperature is
higher than 0.97 at these stations (Table 3). For annual soil temperature (Table 3), R is
generally greater than 0.80 and RMSE is generally less than 1.5 °C for all stations except
for Nangqian where RMSE is 4.51 °C and Lenghu where R is 0.53. These analyses
suggest that the modified LPJ model is able to simulate the temporal evolution of the
observed top-layer soil temperature on the NTP with reasonable accuracy.

For monthly soil moisture, the simulations are largely consistent with the observations in
terms of magnitude and seasonal cycles as reflected by RMSE and R in the range of 0.08
- 0.14 m$^3$/m$^3$ and 0.71 – 0.83, respectively, based on limited observations (Table 4).
Slight overestimation of monthly soil moisture is noted at D66 and MS3608 (Table 4).

Dominant PFTs on the NTP during 1957-2009 simulated by the modified LPJ model is
shown in Fig. 4. The simulated spatial patterns of major PFTs such as perennial alpine
meadow, perennial alpine steppe, barren/sparse grassland and temperate needleleaf
evergreen trees compare favorably with the eco-geographic maps from Zheng et al. (2008)
and the vegetation maps of China at 1:1,000,000 scale by CAS (2007).  It should be noted
here that LPJ not only identifies dominant PFTs of each eco-geographic zone but also
creates more diverse PFTs in each zone than those shown by the eco-geographic maps, as





can be expected under the complex terrain and diverse climatic conditions of the study
region. Large discrepancies between the simulated and surveyed PFTs exist in the
northeast and southeast of the NTP. For example, the northeastern NTP is dominated by
temperate semi-arid plateau coniferous forest and steppe (HIIC1) according to Zheng et
al. (2008), while it is characterized by alpine meadow mixed with temperate needleleaf
evergreen forests, perennial temperate summergreen scrub/grassland, and alpine steppe in
model simulations. Overall, temperate needleleaf evergreen forest (TNEG hereafter),
perennial alpine meadow (PAMD), perennial alpine steppe (PASP), perennial temperate
summergreen shrub/grassland (TSGS), and barren/sparse grassland prevail over the NTP.

3.2 Changes in FPCs and climatic factors
The Mann-Kendall trends of annual total FPC (the sum of FPCs of all PFTs in one grid
cell), top layer annual soil moisture and temperature, annual precipitation and air
temperature during 1957-2009 are presented in Fig. 5. For FPC, 34% (13%) of the region
shows increasing (decreasing) trends. Decreasing FPCs are found mostly in the northwest
(barren/sparse grassland) and east (TSGS) of the NTP, while increasing FPCs are located
mainly in the northeast and southwest where alpine meadow, steppe and temperate
summergreen shrub/grassland dominate. These change patterns in FPCs are largely
consistent with those of NDVI (Zhong et al., 2010) and NPP (Piao et al., 2012) on the
NTP, further demonstrating LPJ's ability in satisfactorily simulating FPCs and their
changes.





Precipitation increases significantly in the northeast but decreases in the northwest and
east of the NTP over the last five decades (Fig. 5). This change pattern in precipitation
largely resembles that of total FPC. Annual changes in the top layer soil moisture also
show a similar spatial pattern to that of precipitation although the trends in soil moisture
are generally small over 79% of the region. Both the top layer soil temperature and air
temperature exhibit warming trends over the entire NTP, with significant trends in the
northwest and the east (Fig. 5). Hence, compared to increasing temperatures, changes in
precipitation appear to play a more important role in determining the spatial patterns of
FPC changes on the NTP. However, over the northwestern and eastern NTP, the
decreasing FPC trends may also be influenced by the warming in addition to the
decreases in precipitation.

The Mann-Kendall trends of FPCs of the four dominant vegetation types, TNEG, PAMD,
PASP and TSGS, are shown in Fig. 6. TNEG displays patches of increasing FPCs (13%
of the entire area) in the northeast and southeast of the NTP. PAMD (PASP) exhibits
predominantly increasing (decreasing) FPCs within the Qinghai Province, accounting for
45% (44%) of the entire area. FPCs of TSGS increase (13% of the entire area) mainly in
the northeast and decrease (9% of the entire area) mainly in the southeast of the NTP.
Overall, it appears that PAMD has invaded into the domain of PASP over the past 50
years.

To further investigate possible vegetation migration caused by climate change over the
NTP during 1957-2009, we examine FPC differences between simulations with and



without the historical trends in climate variables retained (i.e., Historical – S6 in Table 2).
The results presented in Fig. 7 suggest that by climate change alone, total FPC (Fig. 7a)
would increase by about 20-30% in the northeast and southwest of the NTP but decrease
by less than 30% in some sporadically vegetated locales. FPC decreases in the
northwestern NTP where sparse grassland meets bare land implies an encroachment of
desertification in that region and is especially worrisome. Climate change causes
increases in FPC of TNEG by about 30-60% in most of the eastern NTP (Fig. 7b), and it
decreases FPC of PAMD (30-60%) in the eastern NTP but increases FPC of PAMD (<
30%) in the higher interior of the Qinghai Province, resulting in westward migration of
PAMD (Fig. 7c). On the other hand, as a result of climate change, FPC of PASP
decreases in most of the Qinghai Province but increases (by >30%) in the westernmost
part of the NTP (Fig. 7d). TSGS increases by less than 30% in the northeastern NTP but
decreases by more than 30% in the southeast (Fig. 7e) due to climate change. The spatial
patterns of the FPC changes due to climate change correspond well with those of the FPC
trends (Figs. 5, 6), indicating the dominant role of climate change in governing vegetation
changes and dynamics on the NTP.

3.3 Sensitivity of total FPC to changes in climatic factors
$E_{PRCP}$ is positive in 40% of the area and is often larger than 3, meaning that 10%
precipitation increase could lead to more than 3-fold increase in total FPC in warm and
dry places where alpine meadow, barren/sparse grassland, and temperate summergreen
scrub/grassland grow (Fig. 8a). Isolated negative $E_{PRCP}$ are located mostly in the high
elevation of the southern NTP. About 15% (35%) of the NTP shows positive (negative)



$E_{AT}$. Negative $E_{AT}$ (-3 - -0.5) is mostly found in the northern NTP  (Fig. 8b), indicating
that 1° C warming could lead to 0.5 – 3 fold decrease in total FPC. In the far southwest
(32°-36° N and 90°-93° E) where mean annual air temperature is about -10 °C and where
permafrost soil prevails, $E_{AT}$ is significantly positive, implying that warming could
dramatically increase FPC by more than 4 folds.

1 °C increase in soil temperature could decrease FPC by up to 4 folds in the northern
NTP (Fig. 8c); however, in the meantime, 10% increase in soil moisture would result in
up to 5 folds of increase in FPC in roughly the same area (Figs. 8d), suggesting that FPC
is highly sensitive to soil moisture changes especially in the climatologically dry
northwest. In the south NTP, FPC seems insensitive to changes in either soil temperature
or soil moisture (Figs. 8c, d).

Positive $E_{CO2}$ is slightly more widely distributed than $E_{PRCP}$ and $E_{SM1}$ but $E_{CO2}$ is in
general much smaller in magnitude than $E_{PRCP}$ and $E_{SM1}$ (Fig. 8e). About 62% (4%) of the
cells show positive (negative) $E_{CO2}$. The spatial patterns of elasticity show that foliage
growth in heat or water limited NTP is very sensitive to environmental changes. Figure 8f
depicts the dominant drivers that affect total FPC in each grid cell. Precipitation increase
(light green in Fig. 8f) displays major influence on total FPC in the north with primarily
positive effects (crosses in Fig. 8f). Air temperature increase is less important than
precipitation increase and could exert either positive (crosses in Figs. 8f) or negative
(diamonds in Fig. 8f) effects on total FPC depending on the locations. Generally speaking,
positive (negative) effects due to air temperature increase tend to be clustered in the



relatively cold and wet southwest (dry northwest and warm southeast). $CO_2$ change
impacts are mainly seen over the south and some patchy areas of the north with mixed
positive and negative effects. Compared to the other environmental variables, ST1 and
SM1 do not emerge as the dominant factors for FPC changes, indicating that frozen soil
degradation related to soil temperature and moisture changes is not as important as
changes in precipitation, air temperature and $CO_2$ for FPC.

3.4 Sensitivity of the FPC of individual PFTs to changes in climatic factors
The FPC of TNEG increases by 1.6-fold on average in response to 10% precipitation
increase in the eastern NTP (about 17% of the entire area, Fig. 9a, Table 5). The FPC of
PAMD increases in 51% of the area by more than 1.2-fold in the east, north and south of
the Qinghai Province, but decreases in 5% of the entire area by about 1-fold in the bare
and sparse grassland and by about 5-fold in several cells in the eastern and southern NTP
as precipitation increases by 10%. The FPC of PASP decreases in the northeast and south
of the Qinghai Province (24% of the region, Table 5) by about 1-fold, but increases in the
northwest desert region of the Qinghai Province by 1- to 3-fold (Fig. 9c). More cells
showing positive $E_{PRCP}$ for PAMD than for PASP indicates that precipitation increase
would benefit PAMD more than PASP. It appears that as precipitation increases, PAMD
takes over PASP in many cells while PASP encroaches the desert area. The FPC of TSGS
decreases in the southeast by about 1.8-fold but increases by 1.1-fold in the northeast of
the NTP with 10% precipitation increases (Fig. 9d). The southeast NTP is not water
limited and hence increasing precipitation has negative impacts in general.



Large $E_{AT}$ for TNEG is found primarily in the eastern NTP, with positive (16% of the
cells) and negative (17% of the cells) $E_{AT}$ occurring side by side (Fig. 9e). With 1 °C air
temperature increase, PAMD shows positive $E_{AT}$ (about 5) in the southwest where energy
is limited, but negative $E_{AT}$ (-1 - -5) in the north, east and southeast (Fig. 9f).  For PASP,
large and positive $E_{AT}$ is found predominantly over the westernmost tip of the NTP,
whereas nearly the entire Qinghai Province (59% of the region) corresponds to negative
$E_{AT}$ (Fig. 9g, Table 5), indicating that PASP would decline in general as air temperature
increases. TSGS shows mixed positive and negative $E_{AT}$ primarily in the northeast and
southeast, respectively (Fig. 9h). For TSGS and TNEG, $E_{AT}$ is close to zero over nearly
the entire Qinghai Province (Fig. 9, Table 5), because of the bioclimatic restriction of
their establishment.

Although soil temperature and moisture changes do not contribute significantly to total
FPC changes (Fig. 8f), they do affect individual PFTs in a nearly opposite way which
may have given rise to some cancellation in FPC changes. For example, with the
exception of TNEG, negative $E_{ST1}$ over the north and positive $E_{ST1}$ over the southeast for
PAMD, PASP and TSGS correspond respectively to positive and negative $E_{SM1}$ in the
same areas (Fig. 10), although $E_{SM1}$ is generally larger in magnitude than $E_{ST1}$. For TNEG,
slightly negative $E_{ST1}$ is located over the east but highly positive $E_{SM1}$ is seen over the
north (Figs. 10a, 10e). Compared to $E_{AT}$, $E_{ST1}$ is smaller and varies less spatially (Figs.
10a-d). The majority cells (72-83%) display zero $E_{ST1}$ for all four PFTs (Table 5),
indicating that soil temperature is not a sensitive element for foliage growth. However,



soil moisture increase could reduce the coverage of desert, evidenced by the increase of
FPCs of TNEG, PAMD and PASP in the northwest where desert vegetation dominates.

$E_{CO2}$ (Figs. 11a-d) exhibits a similar pattern to that of $E_{PRCP}$ for all four PFTs (Figs. 9a-d).
The numbers of the grid cells with positive, negative and negligible values of $E_{CO2}$ and
$E_{PRCP}$ are also identical for each PFT (Table 5). This similarity between $E_{CO2}$ and $E_{PRCP}$
suggests a strong coupling between photosynthesis and water availability on the NTP.

**4. Discussions**
Our analyses suggest that total FPC changes on the NTP are driven by different
mechanisms over different regions. For example, the increases of total FPC in the
southwest during 1957-2009 identified in Figs. 5 and 7a are due to warming induced
increases in alpine meadow and steppe. Over the northeast of the NTP, changes in total
FPC are determined by the balance between precipitation, soil moisture and $CO_2$ increase
induced expansion (contraction) of temperate needleleaf evergreen forest, perennial
alpine meadow, perennial temperate summergreen scrub/grassland (perennial alpine
steppe) and warming induced decreases in all FPCs. Decreases of total FPC in the
northwestern NTP are related to the negative effects of warming and drying (Fig. 5) on
the growth of alpine meadow and steppe which apparently overwhelms the positive
effects of $CO_2$ increase. In the southeast, changes in total FPC are generally small, likely
because of the thriving TNEG growth induced by the increase of temperature,
precipitation and $CO_2$ cancelled by the decline of PAMD because of warming, and
decreased TSGS due to wetting and $CO_2$ increase. Similarly, in the central region, PAMD



and PASP respond oppositely to the changes in precipitation, air temperature, soil
moisture and $CO_2$, and as a result total FPC shows little change.

Different regions of the NTP are characterized by distinctive climatic features, and hence
vegetation growth in those regions is limited by varying climatic factors. For example,
warming and wetting in the southwest of the NTP make it more suitable for alpine
meadow and steppe to grow. On the other hand, the northwestern NTP has very limited
annual precipitation (<100 mm), and the warming could make it even drier (as observed
during the recent decades, see Fig. 5), posing an increasing challenge for plant growth.
As climate changes, bioclimatic zones will change accordingly.

Another noteworthy finding is that as soil temperature increases across the region, there
are more grid cells showing decreasing (14%) than increasing (5%) top layer annual soil
moisture (Fig. 5). The rise in soil temperature on the NTP increases liquid soil moisture
during cold months because of the increased soil thawing but decreases liquid soil
moisture during warm months because of the enhanced soil evaporation in shallow soil
layers (Cuo et al., 2015). Clearly, the decrease in top layer soil moisture in the warm
growing season could negatively impact vegetation growth in the already dry area and
could accelerate desertification unless the lost moisture can be replenished by increasing
precipitation. In the northern NTP, the negative effects of top layer soil temperature
increases on vegetation growth may also serve as an indication of the consequences of
frozen soil degradation that is happening on the NTP (Cuo et al., 2015).





On the NTP, decreased (increased) vegetation growth in the northwest (southwest and
northeast) will result in reduction (enhancement) in roughness length and increase
(decrease) in albedo, changes in stomatal resistance, etc. These changes in biogeophysical
properties over the region will feedback to the momentum and carbon exchange, water
and energy balances and will undoubtedly affect large scale circulations such as the onset
and intensity of South and East Asia monsoons (Wu et al., 2007; Shi & Liang, 2014; Cui
et al., 2015; He et al., 2015), thereby affecting the regional and global climate.

To the authors' knowledge, this work is the first of its kind in that a state-of-the-art
dynamic vegetation model is applied over the NTP for examining the impacts of both
atmospheric conditions and soil physical conditions on plant coverages, and shows that
atmospheric conditions dominate over the soil physical conditions in affecting the FPC
change. This is highly relevant and timely given the fact that the Tibetan Plateau is
experiencing warming and frozen soil degradation. Also, the output of time series
vegetation type maps can be used in hydrological models to further investigate land cover
change impacts on hydrological processes in the region where major Asian rivers are
originated but where such long term time series land cover maps do not exist. Clearly,
understanding the vegetation changes and the underlying mechanisms over the TP is the
first step towards an understanding of the change impacts of TP's surface conditions on
water resources, hydrological cycles and climate at regional and global scales.

In this study, the role of $CO_2$ in FPC changes is discussed solely in the context of
photosynthesis. However, $CO_2$ is a greenhouse gas and increasing $CO_2$ concentrations





have been credited as one of the primary driving forces behind the global warming.
Without utilizing a fully coupled dynamic atmosphere-land surface-vegetation model it
appears to be rather difficult to separate the effects of $CO_2$ between photosynthesis
related and greenhouse gas related.

**5. Conclusions**
In summary, this study documents the changes in PFTs represented by FPCs on the NTP
during the past five decades and the possible mechanisms behind those changes through
examining the responses of PFTs to changes in climate variables of precipitation, air
temperature, atmospheric $CO_2$ concentrations, 40-cm-deep soil temperature and moisture.
Among the five variables, precipitation is found to be the major factor influencing the
total vegetation coverage positively, while root zone soil temperature is the least
important one with negative impacts. About 34% of the NTP exhibits increasing total
FPC trends compared to 13% with decreasing trends during 1957-2009. Individual PFTs
respond differently to the changes in the five climate variables. The different responses of
individual PFTs to climate change give rise to spatially varying patterns of vegetation
change. Spatially diversified changes in vegetation coverage on the NTP are the result of
changes in heterogeneous climatic conditions in the region, competitions among various
PFTs for energy and water, and regional climate-determined bioclimatic restrictions for
the establishment of different PFTs. The effects of the climate change induced regional
plant functional type changes on water resources and hydrological cycles in one of the
world's largest and most important headwater regions, on the partition of sensible and
latent heat fluxes, and hence on the onset and intensity of south and east Asian monsoon



circulations should be examined further.

**Acknowledgement**
This study is supported by the National Basic Research Program (grant 2013CB956004),
by the National Natural Science Foundation of China (grant 41190083), and by the
Hundred Talent Program granted to Lan Cuo by the Chinese Academy of Sciences. The
National Center for Atmospheric Research (NCAR)'s Advanced Study Program (ASP) is
also acknowledged for providing partial funding for this work.

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





Table 1. Calibrated parameters of plant functional types in the Lund-Potsdam-Jena DGVM model.

| Plant functional types | $g_{min}$ (mm/s) | GDDs | $GDD_{5min}$ | Int | $W_{s-m}$ | $RD_0$ | $TLCO_2$ (°C) | $TUCO_2$ (°C) | TLP (°C) | TUP (°C) | $TL_{cold}$ (°C) |
|---|---|---|---|---|---|---|---|---|---|---|---|
| Temperate broadleaf evergreen | 1.6 | 400 | 900 | 2.5 | 0.2 | 0.8 | -3 | 30 | 15 | 30 | -0.1 |
| Temperate needleleaf evergreen | 1.8 | 300 | 600 | 2.7 | 0.4 | 0.8 | -10 | 15 | 5 | -1 | -15.5 |
| Temperate broadleaf summergreen | 1.5 | 300 | 700 | 1.0 | 0.2 | 0.8 | -3 | 20 | 5 | 20 | -10 |
| Perennial alpine meadow | 0.05 | 70 | 60 | 0.1 | 0.2 | 1.0 | -6 | 15 | 0 | 15 | -20 |
| Perennial alpine steppe | 0.05 | 10 | 20 | 0.1 | 0.2 | 1.0 | -20 | 5 | -7 | 5 | -25 |
| Temperate scrub grass | 0.4 | 200 | 500 | 0.5 | 0.2 | 1.0 | -10 | 18 | 5 | 18 | -10 |

Note: $g_{min}$: minimum canopy conductance; GDDs: number of growing degree days to attain full leaf cover; $GDD_{5min}$: 5 °C based minimum degree day; Int:
interception storage, $W_{s-m}$: water scalar value at which leaves shed by drought for deciduous plant; $RD_0$: fraction of roots in the upper soil layer (0-40 cm);
$TLCO_2$: lower temperature limit for $CO_2$; $TUCO_2$: upper temperature limit for $CO_2$; TLP: lower temperature limit for photosynthesis; TUP: upper temperature
limit for photosynthesis; $TL_{cold}$: lower limit of the coldest monthly mean temperature.
Table 2. Scenarios used for examining the FPC sensitivity to climate elements.

| Scenarios | Variables changed | Changed amount |
|---|---|---|
| S1 | 40-cm-deep daily soil temperature (ST1) | +1 °C |
| S2 | 40-cm-deep daily soil moisture (SM1) | +10% |
| S3 | Monthly air temperature (AT) | +1 °C |
| S4 | Monthly precipitation (PRCP) | +10% |
| S5 | Annual $CO_2$ | +10% |
| S6 | AT, PRCP, wet day, $CO_2$ | Trends removed |
| Historical | - | - |



Table 3. Statistics of the observed and simulated monthly and annual mean soil
temperature in 0-40 cm depth. The observations end in 2009 for all stations except
for Wudaoliang and Xidatan for which the observations end in 2006. R: correlation
coefficient; RMSE: root mean square error.

| Stations | Start | Latitude | Longitude | Elevation (m) | Obs. (°C) | Sim. (°C) | R | RMSE (°C) |
|---|---|---|---|---|---|---|---|---|
| | | | Monthly | | | | | |
| Wudaoliang | 2005 | 35.217 | 93.083 | 4612.2 | -1.74 | -1.57 | 0.96 | 2.89 |
| Maduo | 2004 | 34.917 | 98.217 | 4272.3 | 2.13 | 0.91 | 0.99 | 1.40 |
| Mengyuan | 2004 | 37.383 | 101.617 | 2938.0 | 5.29 | 6.05 | 0.99 | 1.67 |
| Mangai | 2004 | 38.250 | 90.850 | 2944.8 | 8.81 | 9.70 | 1.00 | 2.68 |
| Lenghu | 2004 | 38.750 | 93.333 | 2770.0 | 7.50 | 6.99 | 1.00 | 3.07 |
| Xinghai | 2004 | 35.583 | 99.983 | 3323.2 | 5.71 | 5.27 | 0.99 | 1.40 |
| Zaduo | 2004 | 32.900 | 95.300 | 4066.4 | 5.67 | 5.31 | 0.99 | 1.32 |
| Qilian | 2004 | 38.183 | 100.250 | 2787.4 | 5.88 | 4.70 | 1.00 | 2.08 |
| Xidatan | 2005 | 35.717 | 94.133 | 4538.0 | -0.45 | -0.70 | 0.98 | 3.21 |
| Golmud | 2004 | 36.417 | 94.900 | 2807.6 | 9.16 | 11.89 | 0.99 | 3.03 |
| | | | Annual | | | | | |
| Mangai | 1989 | 38.250 | 90.850 | 2944.8 | 8.17 | 8.15 | 0.82 | 0.53 |
| Lenghu | 1981 | 38.750 | 93.333 | 2770.0 | 7.56 | 7.82 | 0.53 | 0.70 |
| Delingha | 1982 | 37.367 | 97.367 | 2981.5 | 7.23 | 7.85 | 0.94 | 0.67 |
| Gangcha | 1981 | 37.333 | 100.133 | 3345.0 | 3.59 | 3.89 | 0.90 | 0.38 |
| Mengyuan | 1984 | 37.383 | 101.617 | 2938.0 | 4.69 | 5.63 | 0.93 | 0.97 |
| Germud | 1977 | 36.417 | 94.900 | 2807.6 | 8.53 | 10.48 | 0.68 | 2.08 |
| Qiabuqia | 1983 | 36.267 | 100.617 | 2835.0 | 7.89 | 8.23 | 0.82 | 0.67 |
| Xining | 1962 | 36.717 | 101.750 | 2295.2 | 9.02 | 9.73 | 0.67 | 0.90 |
| Minhe | 1994 | 36.317 | 102.850 | 1813.9 | 11.22 | 11.57 | 0.61 | 0.74 |
| Xinghai | 1993 | 35.583 | 99.983 | 3323.2 | 5.48 | 5.23 | 0.80 | 0.36 |
| Qumalai | 1984 | 34.133 | 95.783 | 4175.0 | 3.05 | 1.46 | 0.90 | 1.63 |
| Maduo | 1981 | 34.917 | 98.217 | 4272.3 | 1.47 | 0.88 | 0.87 | 0.75 |
| Dari | 1981 | 33.750 | 99.650 | 3967.5 | 3.03 | 2.20 | 0.87 | 0.90 |
| Henan | 1982 | 34.733 | 101.600 | 3670.0 | 3.85 | 3.83 | 0.90 | 0.79 |
| Jiuzhi | 1979 | 33.433 | 101.483 | 3628.5 | 4.58 | 3.84 | 0.90 | 0.79 |
| Nangqian | 1994 | 32.200 | 96.483 | 3643.7 | 8.63 | 4.13 | 0.92 | 4.51 |


Table 4. Statistics of the first layer (0-40 cm) monthly soil moisture. The observation
period is August 1997 – September 1998.

| Stations | Latitude | Longitude | Elev. (m) | Mean Obs. (m³/m³) | Mean Sim. (m³/m³) | R | RMSE (m³/m³) |
|---|---|---|---|---|---|---|---|
| Amdo | 32.25 | 91.63 | 4700 | 0.14 | 0.15 | 0.76 | 0.10 |
| D66 | 35.52 | 93.78 | 4600 | 0.08 | 0.13 | 0.83 | 0.10 |
| MS3608 | 31.24 | 91.78 | 4610 | 0.16 | 0.20 | 0.80 | 0.14 |
| Tuotuohe | 34.22 | 92.43 | 4353 | 0.12 | 0.13 | 0.71 | 0.08 |

Table 5. Numbers of the cells that display positive (+), negative (-) and non or
negligible (n) elasticity for the four major plant functional types with precipitation



increased by 10% (Prcp 10%), air temperature increased by 1 °C (AT+1),  top layer
soil temperature increased by 1 °C (ST+1), top layer soil moisture increased by 10%
(SM 10%), and $CO_2$ concentrations increased by 10% ($CO_2$ 10%). There are 2052
grid cells in total. TNEG: temperate needleleaf evergreen, PAMD: perennial alpine
meadow, PASP: perennial alpine steppe, TSGS: temperate summer green
grass/shrub

| Scenarios | Cell signs | TNEG | PAMD | PASP | TSGS |
|---|---|---|---|---|---|
| Prcp 10% | + | 368 | 1064 | 530 | 184 |
| | - | 68 | 112 | 509 | 317 |
| | n | 1616 | 876 | 1013 | 1551 |
| AT+1 | + | 216 | 761 | 132 | 348 |
| | - | 364 | 711 | 1226 | 237 |
| | n | 1472 | 580 | 694 | 1467 |
| ST+1 | + | 19 | 39 | 2 | 254 |
| | - | 327 | 522 | 421 | 165 |
| | n | 1706 | 1491 | 1629 | 1633 |
| SM 10% | + | 701 | 1012 | 418 | 142 |
| | - | 36 | 72 | 492 | 340 |
| | n | 1315 | 968 | 1142 | 1570 |
| $CO_2$ 10% | + | 280 | 1302 | 590 | 185 |
| | - | 99 | 64 | 557 | 336 |
| | n | 1673 | 686 | 905 | 1531 |




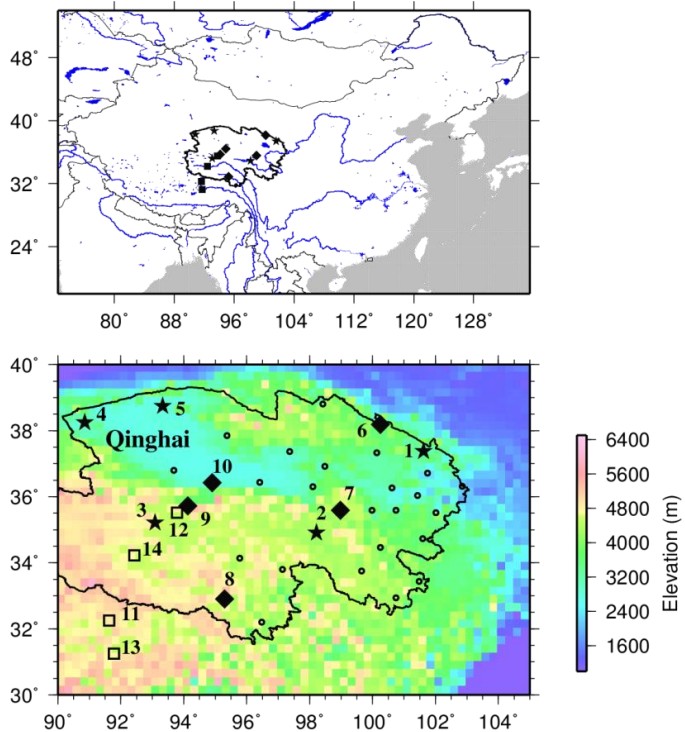

Figure 1. Geographic locations of the study domain and the stations. Black lines
outline the boundary of the Qinghai Province. Stars represent the stations whose
observations are used to develop the linear regression relationships between daily
air temperature and 0-40cm depth daily soil temperature. Stations denoted as
diamonds are for monthly soil temperature evaluation and circles are for annual soil
temperature evaluation.  The stations are: 1: Mengyuan; 2: Maduo; 3: Wudaoliang; 4:
Mangai; 5: Lenghu; 6 Qilian; 7: Xinghai; 8: Zaduo; 9: Xidatan; 10: Germud; 11: Amdo;
12: D66; 13: MS3608; 14: Tuotuohe. Among the stations, Wudaoliang, Xidatan,
Amdo, D66, MS3608 and Tuotuohe are permafrost soil sites and all others are
seasonally frozen soil sites. Stations 1-10 and circles were used for soil temperature
validation while stations 11-14 (empty squares) were used for soil moisture
validation.





Figure 2. Simulated and observed monthly soil temperature at 5 calibration sites.







Figure 3. Simulated and observed monthly soil temperature at 5 evaluation sites.



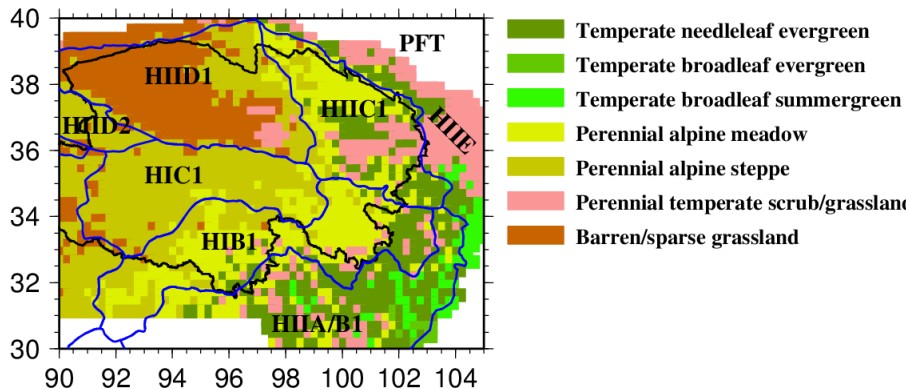

Figure 4. Eco-geographic regions from Zheng et al. (2008) (blue lines) and the LPJ
simulated dominant plant functional types represented by foliar projective covers
(FPCs) under full leaf during 1957-2009. The eco-geographic regions are: HIIC1:
plateau temperate semi-arid high mountain and basin coniferous forest and steppe
region; HIID1: plateau temperate arid desert region; HIID2: plateau temperate arid
desert region; HIC1: plateau sub-cold semi-arid alpine meadow-steppe region; HIB1:
plateau sub-cold sub-humid alpine shrub–meadow region; HIIA/B1: plateau
temperate humid/sub-humid high mountain and deep valley coniferous forest
region; and HIIE: temperate shrub grass-desert region. Black line outlines the
Qinghai Province.





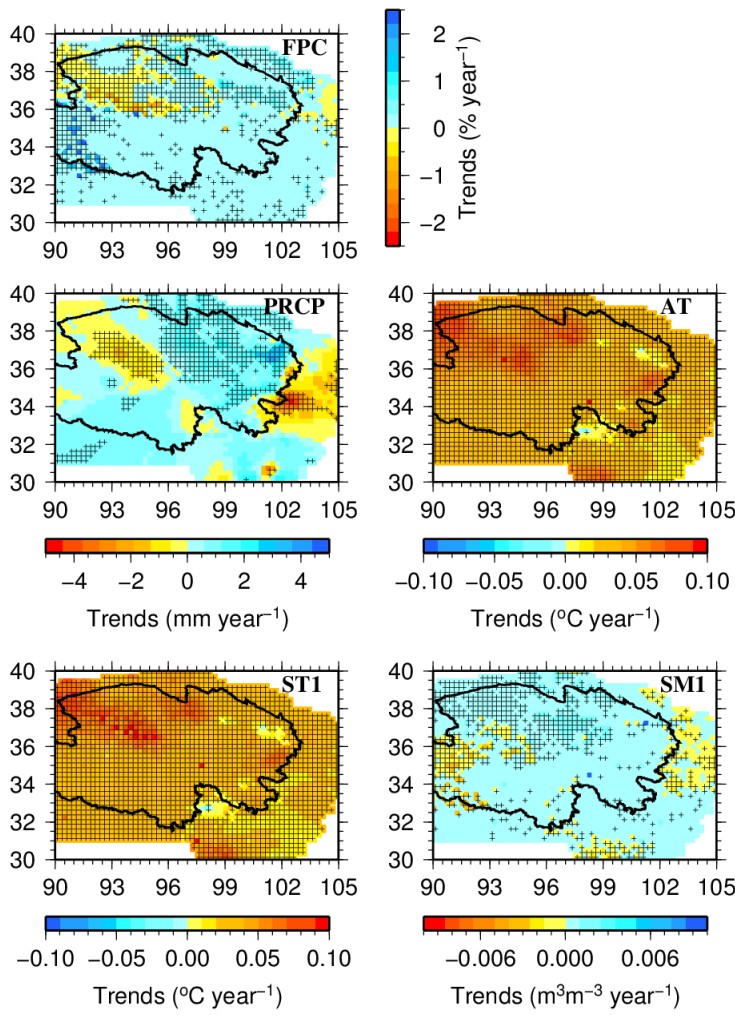

Figure 5. Mann-Kendall trends of simulated annual total FPC, 0-40 cm depth soil
temperature (ST1), 0-40 cm depth soil moisture (SM1), and observed precipitation
(PRCP) and air temperature (AT).





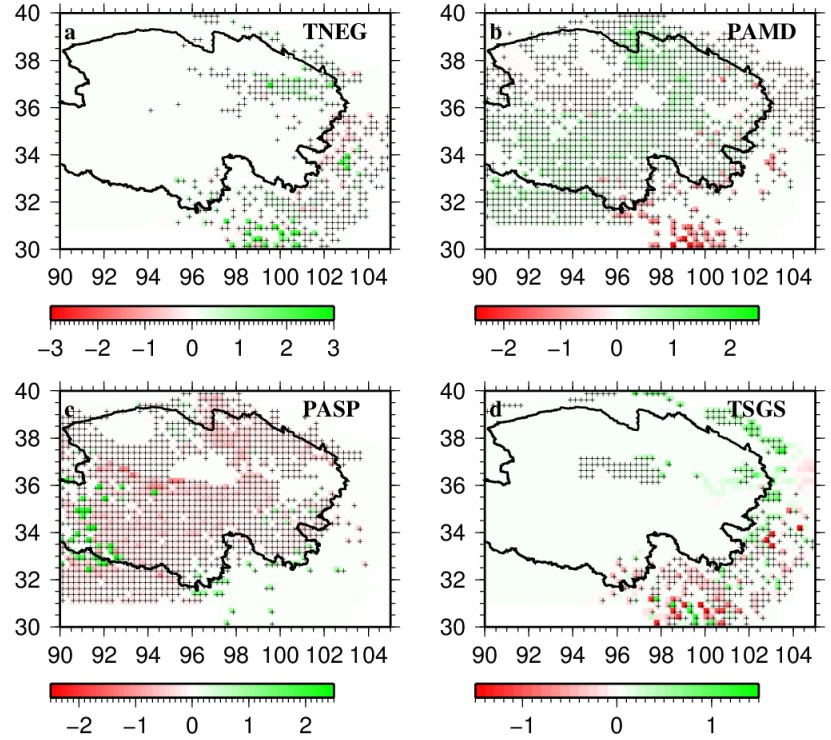

Figure 6. Simulated Mann-Kendall trends of individual FPCs in 1957-2009. TNEG:
Temperate needleleaf evergreen, PAMD: perennial alpine meadow, PASP: perennial
alpine steppe, TSGS: temperate summer green scrub/grassland.





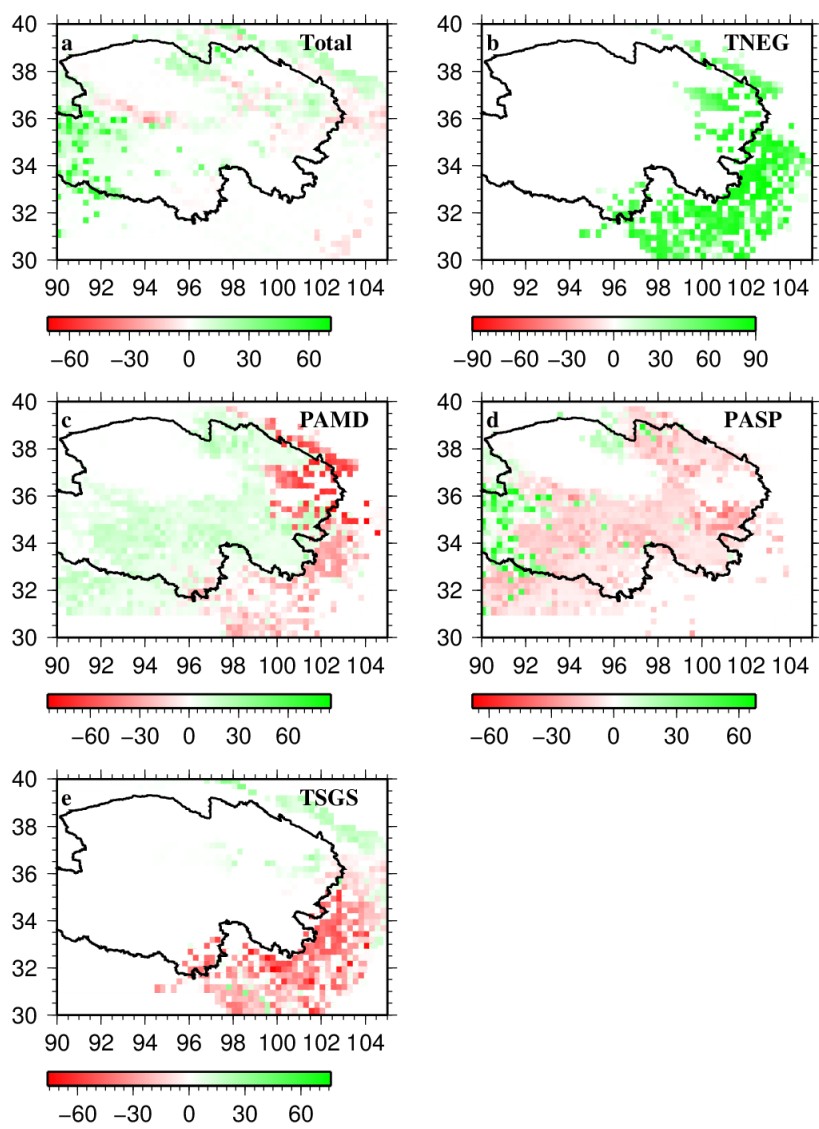

Figure 7.  Differences (%) in total FPC and individual FPCs between historical
climate simulation and trend-removed climate simulation.




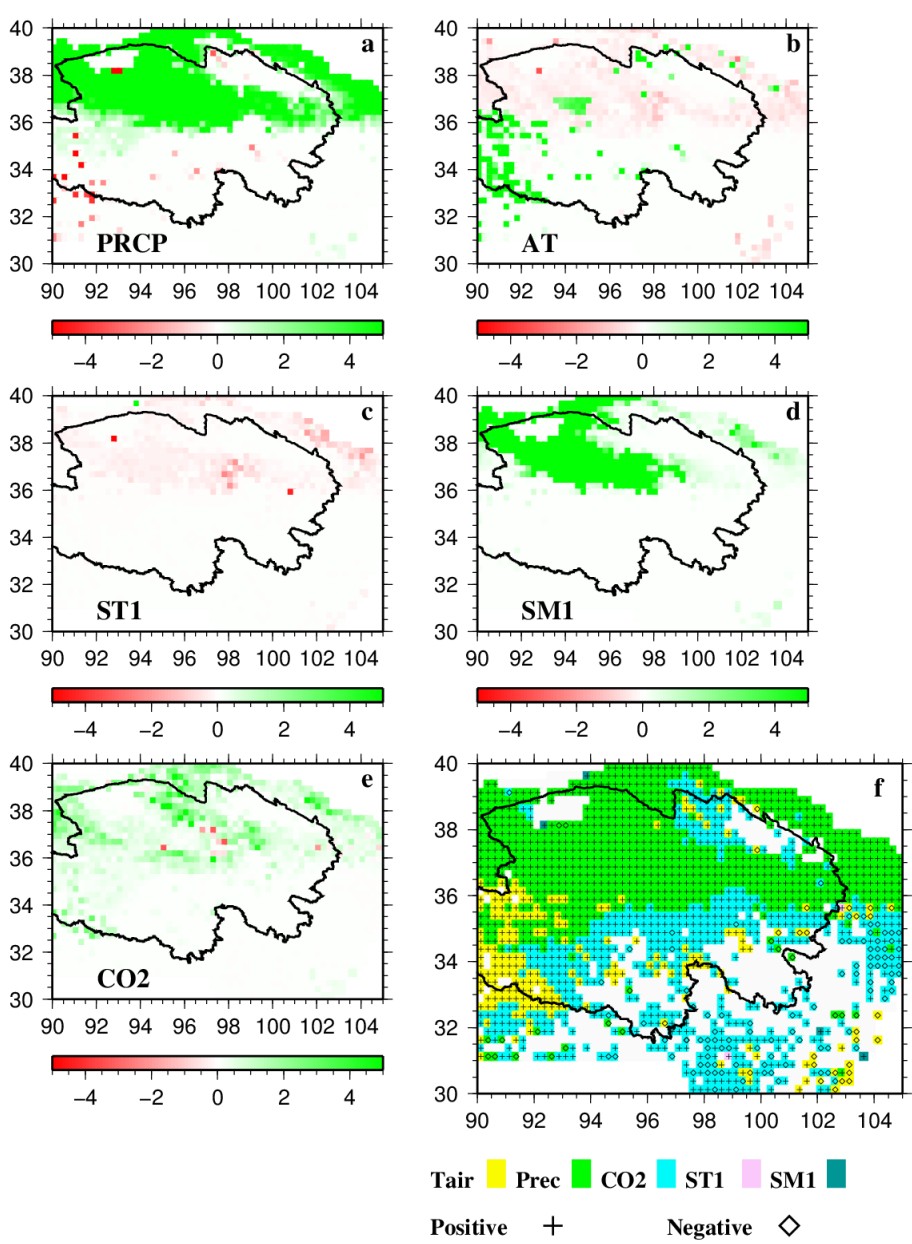

Figure 8. Elasticity of total FPC with precipitation increased by 10% (a), air
temperature increased by 1 °C (b), soil temperature increased by 1 °C (c), soil
moisture increased by 10% (d), $CO_2$ increased by 10% (e), and dominant elasticity
related with changes in precipitation, air temperature, soil temperature, soil
moisture and $CO_2$ (e). In (e), + (plus) and ◊ (diamond) represent positive and
negative elasticity, respectively.



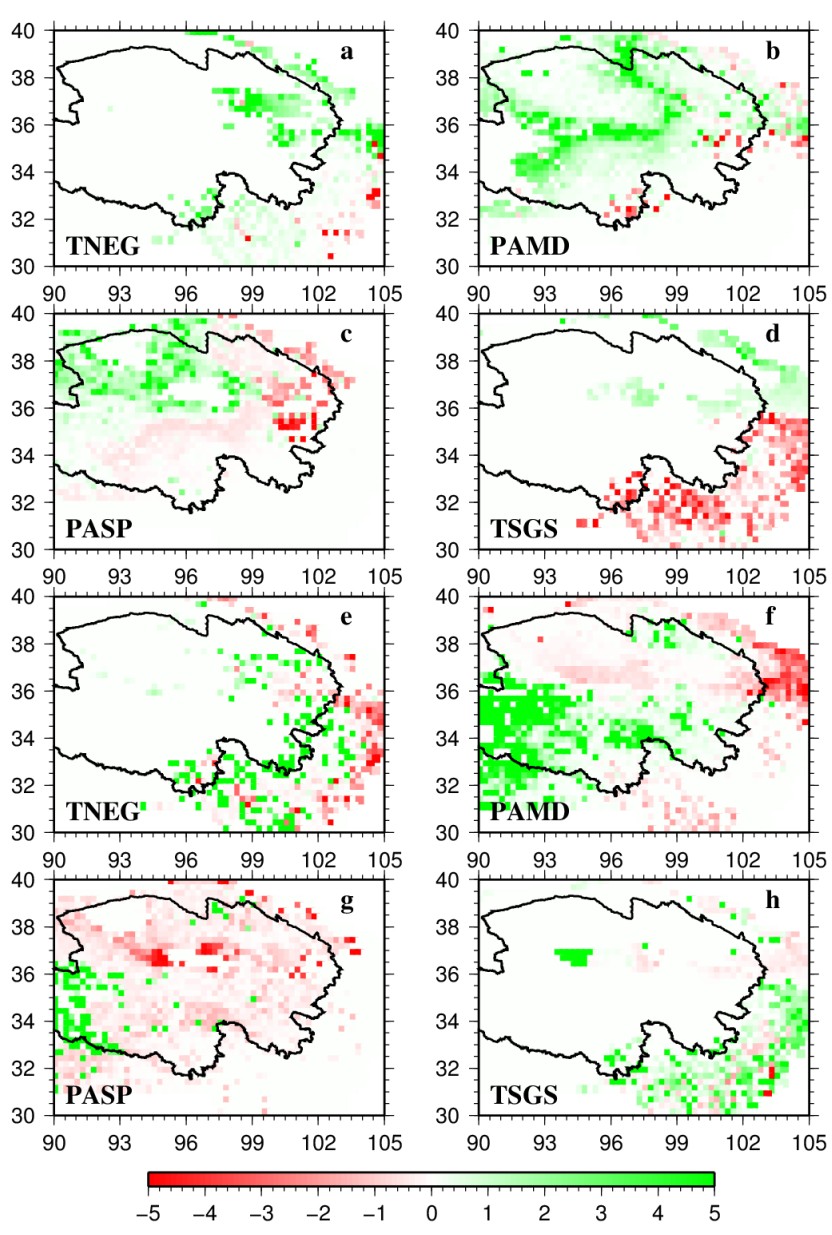

Figure 9. Elasticity of individual FPCs with precipitation increased by 10% (a-d) and
air temperature increased by 1 °C (e-h). TNEG: temperate needleleaf evergreen;
PAMD: perennial alpine meadow; PASP: perennial alpine steppe; TSGS: temperate
summer green scrub/grassland.





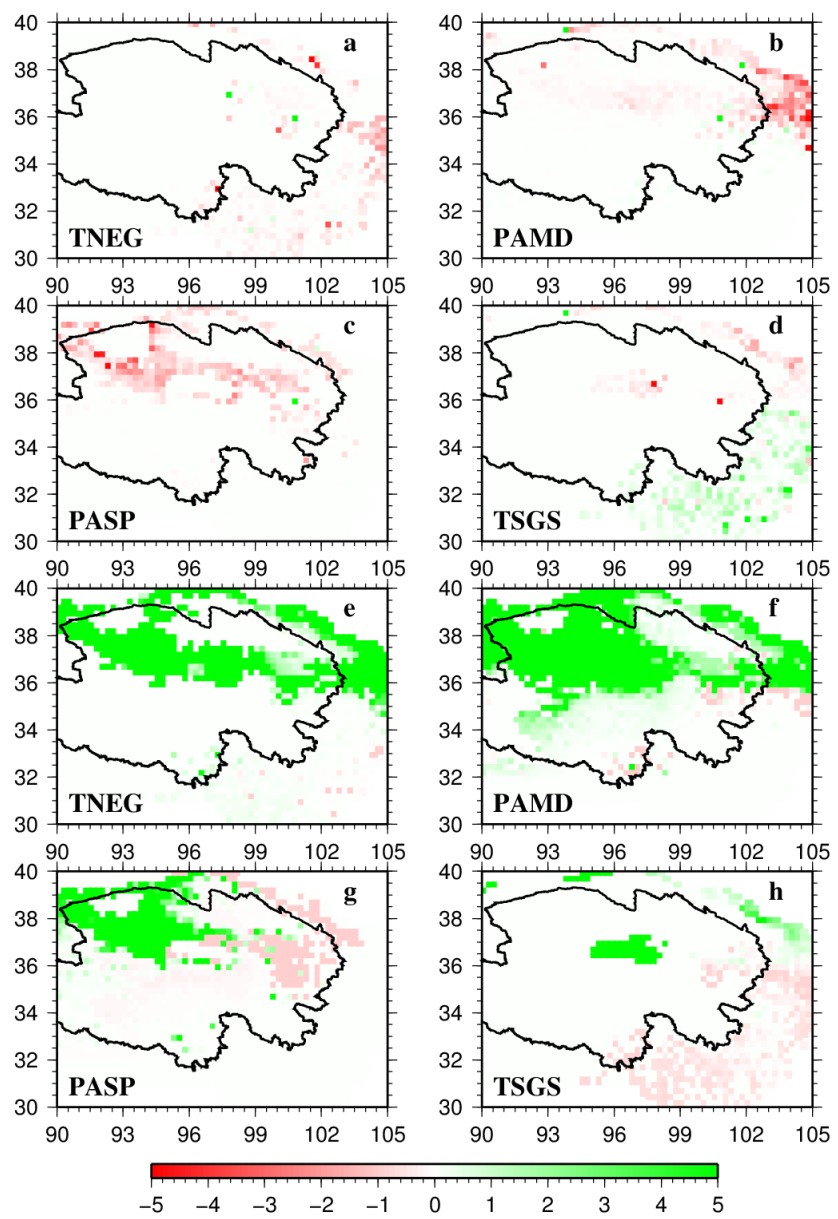

Figure 10. Elasticity of individual FPCs with +1 °C soil temperature increase (a-d)
and 10% soil moisture increase (e-h).





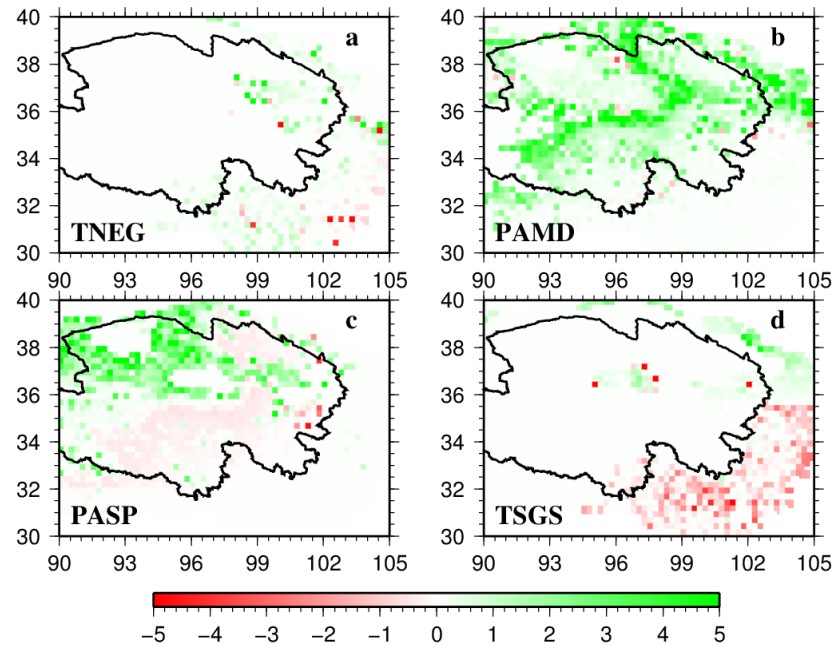

Figure 11. Elasticity of individual FPCs with 10% $CO_2$ increase.