# Peer review of "Annual changes in plant functional types and their responses to climate change on the Northern Tibetan Plateau"

_Biogeosciences, 2016_

## Referee Comment (RC1) · Anonymous Referee #1 · 24 Mar 2016

In the manuscript "Annual changes in plant functional types and their responses to climate change on the northern Tibetan Plateau", Cuo et al. investigated the changes of plant functional types (PFTs) and foliar projective coverages (FPCs) on the northern Tibetan Plateau (NTP) during 1957-2009 with an improved Lund-Potsdam-Jena Dynamic Global Vegetation Model, where the simulation of temperature and moisture of the top soil layer (0-40cm) were modified in order to better account for the repeated freezing and thawing cycles on the NTP. By comparing the model simulations under different designed scenarios, they also examined the responses of PFTs and FPCs to changes in root zone soil temperature, soil moisture, air temperature, precipitation and $CO_2$. According to their results, area experiencing increased FPCs was larger than that

showing decreased FPCs during this period (34% v.s. 13%). Meanwhile, there was an extensive replacement of temperate scrub grass by perennial alpine meadow. Overall, precipitation played a dominant control on the changes, but the dominant drivers and the responses of different PFTs varied regionally, resulting in spatially heterogeneous patterns of vegetation changes on the NTP. Generally, I find the structure and presentation of the manuscript very clear, and appreciate the importance of investigating the changes in PFTs and FPCs on NTP.

However, as necessary model validation, comparisons with observed PFTs and FPCs were weakly presented or missed. The authors compared the modeled spatial patterns of dominant PFTs with observations based on the eco-geographic map compiled by Zheng et al. and the vegetation map of China by CAS, but they did this with the eco-geographic map by visual comparison without anything statistically (Figure 4), whereas with the vegetation map of China, the comparison results were even not presented. Moreover, even though one major analysis was about FPCs, the author didn't carry out any comparison with the observed FPCs to prove the model's reliability in simulating FPCs on NTP. Therefore, I would suggest the authors (1) presenting PFTs comparisons statistically, such as kappa statistic; (2) presenting the results of the comparison with the vegetation map of China; (3) comparing the model simulations with the satellite-based FPCs products.

I also feel disappointed that the entire analysis was based on model simulations, but this is probably a common way for the community of modellers to perform analysis. Therefore, I won't argue more on that. However, I think that it is more accurate to add "model-based" at the beginning of the manuscript's title, and including more discussions with observation-based studies can certainly spicy up the manuscript. For example, rather than simply saying that the simulated change patterns in FPCs were largely consistent with previous study on NDVI and NPP (Line 382-385), the authors could have more detailed descriptions on the trends or responses found in NDVI observations, and discussing what changes in NPP would be expected due to the changes

in their simulated FPCs, whether those expectations are consistent with the existing studies or not.

Minor Comments:

1. Better to keep consistent when using "carbon" (e.g. Line 48, 54, 550) and "C" (e.g. Line79, 146, 176)

2. Line 262: What method is used for the Interpolation?

3. Table 1: According to this table, the upper temperature limit for photosynthesis (TUP) is −1 degree for temperate needleleaf evergreen tree. Such a low temperature doesn't make sense at all. I hope it was a typo, otherwise it would be worrisome because the authors might get the right result for the wrong reason with the parameter calibration.

4. Table 1: Please complete "temperate broadleaf evergreen" as "temperate broadleaf evergreen tree" or "temperature broadleaf evergreen forest". The same for "temperate needleleaf evergreen" and "temperate broadleaf summergreen". Not only in this table, but also in the legends of figures 4, 6, 9 and 10. What the authors put there is not misleading, but seriously "temperate broadleaf evergreen" is not a plant functional type.

5. Table 1: What do you mean by "... temperature limit for CO2"?

6. Table 2: In the last second row, "CO2" should have "2" as subscribe.

7. Figure 2, 3: It reads better, if the station name could be labeled for each panel.

8. Figure 4-11: why there is a white band at the left corner of those figures?

9. Figure 5: What does the symbol "+" mean? Significance? Better to explain it in the figure legend.

10. Line 900: "... (e). In (e), ..." should be "... (f). In (f), ..."

11. Figure 8: In panel e, CO2 should have "2" as subscribe.

---

## Referee Comment (RC2) · Anonymous Referee #2 · 24 Apr 2016

Re: Annual changes in plant functional types and their responses to climate change on the Northern Tibetan Plateau (by Cuo et al.)

General Comments: The manuscript by L. Cuo describes a new application of well-established dynamics vegetation model of LPJ-DGVM and its response to historical climate change in on the Northern Tibetan Plateau (NTP), which presents a new analysis and advanced quantitative understanding of the effects of changes in precipitation, air temperature, $CO_2$ concentration, soil temperature and soil moisture on plant functional types (via the changes in foliar projective coverages (FPCs)). They found that changes in FPCs across the NTP during 1957-2009, with 34% (13%) of the region showing increasing (decreasing) trends. The precipitation is the major controlling fac-

tor of FPC (with a positive impact) , while the soil temperature increase exhibits small but negative impact on FPCs. The key findings of the paper are of potential interest to global change research community and well fill in scope of the journal. The paper is generally well structured and written and did a reasonable job on presenting the model modifications and calibration and key results.

However, I have two major concerns: 1) lacking a robust model validation for FPC (or FPT). The model validation for soil temperature seems acceptable and reasonable (Fig. 3), but, for the FPC, it seems very weak although the authors tried to simply compare the simulated PFT with a published national vegetation map of China (by Zheng et al, 2008). I am not clear how this has been done and how good is the comparison? I would suggest to use a better quantitative method like Kappa Statistic (Cohen, 1960) to improve the comparsion; Cohen, Jacob (1960). A coefficient of agreement for nominal scales. Educational and Psychological Measurement 20 (1): 37–46.

2) lacking some discussions on the uncertainties of model key parameters (like "CO2 fertilization effect" for 6 main PFT)? Because the CO2 fertilization effect at PFT level seems a big challenge and remain large uncertainty in model simulations. It is not clear for me how the authors have considered and quantify the different response of 6 PFT to increasing CO2 concentration over past 52 years in the model simulations? Did they use the same or different parameters?

Specific points: 1) On page 13, line 276-279: I am not clear this mentioned comparison here. Please clarify this point by providing more information or explanations. 2) On page 14, line 311-317, Why you used the elasticity (E)? What is advantage of the E? Please also provide a key reference if possible. 3) On page 17, line 382-385: It is too general statement here. How did you judge "it is largely consistent"? It would be better to use some quantitative indices or methods to justify this critical point here. 4) For the Fig. 2 -3, There are missing some important information for the figure captions. I have no idea which line is model simulation and which is the observation? Please add this missing information directly in both figures.

[Figure]

I would be happy to see this paper to be accepted after a major revision.

---

## Author Comment (AC1) · 20 May 2016

Reviewer #1
In the manuscript "Annual changes in plant functional types and their responses to climate change on the northern Tibetan Plateau", Cuo et al. investigated the changes of plant functional types (PFTs) and foliar projective coverages (FPCs) on the northern Tibetan Plateau (NTP) during 1957-2009 with an improved Lund-Potsdam-Jena Dynamic Global Vegetation Model, where the simulation of temperature and moisture of the top soil layer (0-40cm) were modified in order to better account for the repeated freezing and thawing cycles on the NTP. By comparing the model simulations under different designed scenarios, they also examined the responses of PFTs and FPCs to changes in root zone soil temperature, soil moisture, air temperature, precipitation and $CO_2$. According to their results, area experiencing increased FPCs was larger than that showing decreased FPCs during this period (34% v.s. 13%). Meanwhile, there was an extensive replacement of temperate scrub grass by perennial alpine meadow. Overall, precipitation played a dominant control on the changes, but the dominant drivers and the responses of different PFTs varied regionally, resulting in spatially heterogeneous patterns of vegetation changes on the NTP.

Generally, I find the structure and presentation of the manuscript very clear, and appreciate the importance of investigating the changes in PFTs and FPCs on NTP. However, as necessary model validation, comparisons with observed PFTs and FPCs were weakly presented or missed. The authors compared the modeled spatial patterns of dominant PFTs with observations based on the eco-geographic map compiled by Zheng et al. and the vegetation map of China by CAS, but they did this with the ecogeographic map by visual comparison without anything statistically (Figure 4), whereas with the vegetation map of China, the comparison results were even not presented. Moreover, even though one major analysis was about FPCs, the author didn't carry out any comparison with the observed FPCs to prove the model's reliability in simulating FPCs on NTP. Therefore, I would suggest the authors (1) presenting PFTs comparisons statistically, such as kappa statistic; (2) presenting the results of the comparison with the vegetation map of China; (3) comparing the model simulations with the satellite based FPCs products.

**Response**: We greatly appreciate the reviewer's thoughtful comments and suggestions. We agree with both reviewers that the evaluation of the simulated FPTs and FPCs was weak in the original version of the manuscript primarily because there were no detailed data from field campaigns and (or) systematic monitoring on the NTP in the past 5 decades. In the revised version, we included the MODIS LAI data from 2000 through 2009 (2009 was the year when the model simulations were ended). We also presented the results of the comparisons with the vegetation map of China generated in 2001 by CAS.

As much as we would like to present the evaluation results statistically such as using the Kappa statistics, we found that to be rather difficult due to the fact that the vegetation classification systems are different between the observed datasets and the model simulations and any statistical computation would be subject to large uncertainties. Specifically, the land cover classification in Zheng et al. (2008) and CAS (2001) are in polygon format and each polygon contains mixed vegetation classes without any

information of the exact location of each individual class, which renders it impossible to convert from the polygons that represent the mixed vegetation classes as a whole to the model grid cells that represent the mixed individual vegetation classes. For example, in Zheng et al. (2008), the mixed vegetation class in a polygon includes both temperate semi-arid coniferous forest and steppe in the northeast of the Tibetan Plateau (HIIC1) without showing the exact location of the individual vegetation type; whereas the LPJ simulations are more specific about the location of each vegetation type within each grid cell. Therefore, we feel that any statistical comparisons would not be fair and might result in undesirably large discrepancies. In the revised version, we nevertheless presented as many quantitative comparisons as possible.

In the revised version, the comparisons of the LPJ simulations with both Zheng et al. (2008) and the Chinese Academy of Sciences (CAS, 2001) surveyed maps are presented (Figure 4a, b below). The comparisons showed that 69% of the cells are similar between the LPJ simulation and Zheng et al. (2008) while 42% of the cells agree with each other between the LPJ simulations and the CAS (2001). The differences between the LPJ simulations, Zheng et al. (2008) and CAS (2001) lie mostly in the southeast in that the CAS (2001) map exhibits various subtropical vegetation types and with temperate scrub/grassland dominated in the southeast; while the LPJ simulations and Zheng et al. (2008) display temperate needleleaf evergreen trees. On the other hand, the LPJ simulation and the CAS (2001) map show more similarity in the northeast where alpine meadow and temperate scrub/grassland are widely distributed than between the LPJ simulations and Zheng et al. (2008)

The MODIS Terra LAI is analyzed and compared with the LPJ simulated FPC for 2000-2009 (Figure 4c, d below). We also analyzed the MODIS Terra/Aqua combined LAI but the data are available only after 2002 and therefore we focused mainly on the MODIS Terra LAI. The spatial patterns of the MODIS LAI and the LPJ simulated FPC show similarities to some extent. For example, in the northwest, where LAI is low, FPC is also small. Major differences exist mainly in the southwest where FPC is greater than 90% but LAI is less than 0.3, most likely because of the small leaf area coverage but high numbers of individual PFTs in the steppe and meadow dominated regions. The spatial patterns of the LPJ simulated PFT (Figure 4a) and the MODIS LAI (Figure 4c) match quite well general, in that barren/sparse grassland corresponds with LAI less than 0.2; alpine steppe corresponds with LAI in 0.2 – 03; alpine meadow corresponds with LAI in 0.3 - 0.5; and temperate forest and scrub/grassland corresponds with LAI greater than 0.8 These analyses indicate that the LPJ simulations, though not perfect, are reasonable.

[Figure]

Figure 4. Eco-geographic regions from Zheng et al. (2008) (blue lines) and the LPJ simulated dominant plant functional types represented by foliar projective covers (FPCs) under full leaf during 1957-2009 (a); Zheng et al. (2008) and CAS (2001) surveyed maps (b); MODIS Terra LAI and Zheng et al. (2008) maps (c); and LPJ simulated FPC and Zheng et al. (2008) maps (d). The eco-geographic regions are: HIIC1: plateau temperate semi-arid high mountain and basin coniferous forest and steppe region; HIID1: plateau temperate arid desert region; HIID2: plateau temperate arid desert region; HIC1: plateau sub-cold semi-arid alpine meadow-steppe region; HIB1: plateau sub-cold sub-humid alpine shrub–meadow region; HIIA/B1: plateau temperate humid/sub-humid high mountain and deep valley coniferous forest region; and HIIE: temperate shrub grass-desert region. Black line outlines the Qinghai Province.

Chinese Academy of Sciences, 2001, 1:1,000,000 China Vegetation Map, China Science Publishing & Media Ltd.

I also feel disappointed that the entire analysis was based on model simulations, but this is probably a common way for the community of modellers to perform analysis. Therefore, I won't argue more on that. However, I think that it is more accurate to add "model-based" at the beginning of the manuscript's title, and including more discussions with observation-based studies can certainly spicy up the manuscript. For example, rather than simply saying that the simulated change patterns in FPCs were largely consistent with previous study on NDVI and NPP (Line 382-385), the authors could have more detailed descriptions on the trends or responses found in NDVI observations, and discussing what changes in NPP would be expected due to the changes in their simulated FPCs, whether those expectations are consistent with the existing studies or not.

**Response**: Thanks for the suggestions. The title of the paper has been changed to "Annual changes in plant functional types and their responses to climate change on the Northern Tibetan Plateau simulated by a dynamic vegetation model". To include more discussions with observation-based studies, we improved lines 382-385 with the following: "The variation in the change was also found by Zhong et al. (2010) who reported that 50% of the entire TP had increased NDVI with 30% of the region had decreased NDVI during 1998-2006, with most of the increases occurring in the alpine steppe and alpine meadow in the TP. Further, the LPJ simulated Mann-Kendall trends of NPP (Figure A below, not shown in the revised manuscript) exhibit similar spatial patterns to those in Piao et al. (2012) in that the increase trends prevail in the northeast and the south of the NTP and more widely spread than those of the total FPC."

[Figure]

Figure A The LPJ simulated annual NPP trends in the NTP.

Minor Comments:

1. Better to keep consistent when using "carbon" (e.g. Line 48, 54, 550) and "C" (e.g. Line79, 146, 176)
**Response**: We use carbon throughout the manuscript.

2. Line 262: What method is used for the Interpolation?
**Response**: Cubic convolution, by which a new value is determined based on fitting a smooth curve through the 16 nearest input cell values. See line 263.

3. Table 1: According to this table, the upper temperature limit for photosynthesis (TUP) is −1 degree for temperate needleleaf evergreen tree. Such a low temperature doesn't make sense at all. I hope it was a typo, otherwise it would be worrisome because the authors might get the right result for the wrong reason with the parameter calibration.
**Response**: Sorry, it was a typo and it should be 20 °C. This has been changed in the revised manuscript.

4. Table 1: Please complete "temperate broadleaf evergreen" as "temperate broadleaf evergreen tree" or "temperature broadleaf evergreen forest". The same for "temperate needleleaf evergreen" and "temperate broadleaf summergreen". Not only in this table, but also in the legends of figures 4, 6, 9 and 10. What the authors put there is not misleading, but seriously "temperate broadleaf evergreen" is not a plant functional type.
**Response**: Thanks. Changed accordingly.

5. Table 1: What do you mean by ": : : temperature limit for CO2"?
**Response**: It is the temperature limit for $CO_2$ absorption. Changed accordingly.

6. Table 2: In the last second row, "CO2" should have "2" as subscribe.
**Response**: Changed accordingly.

7. Figure 2, 3: It reads better, if the station name could be labeled for each panel.
**Response**: Station names are added.

8. Figure 4-11: why there is a white band at the left corner of those figures?
**Response**: The white band region represents high elevation region that was excluded from the analysis as there were not enough meteorological stations for the interpolation of temperature and precipitation.

9. Figure 5: What does the symbol "+" mean? Significance? Better to explain it in the figure legend.
**Response**: "+" means that the trends are statistically significant at 95% confidence level. This text is added in the figure caption.

10. Line 900: ": : : (e). In (e), : : :" should be ": : : (f). In (f), : : :"
**Response**: Changed accordingly. Thanks.

11. Figure 8: In panel e, CO2 should have "2" as subscribe.
**Response**: Changed accordingly.

Reviewer #2

Re: Annual changes in plant functional types and their responses to climate change on the Northern Tibetan Plateau (by Cuo et al.)

General Comments: The manuscript by L. Cuo describes a new application of well established dynamics vegetation model of LPJ-DGVM and its response to historical climate change on the Northern Tibetan Plateau (NTP), which presents a new analysis and advanced quantitative understanding of the effects of changes in precipitation, air temperature, $CO_2$ concentration, soil temperature and soil moisture on plant functional types (via the changes in foliar projective coverages (FPCs)). They found that changes in FPCs across the NTP during 1957-2009, with 34% (13%) of the region showing increasing (decreasing) trends. The precipitation is the major controlling factor of FPC (with a positive impact), while the soil temperature increase exhibits small but negative impact on FPCs. The key findings of the paper are of potential interest to global change research community and well fill in scope of the journal. The paper is generally well structured and written and did a reasonable job on presenting the model modifications and calibration and key results. However, I have two major concerns:

1) lacking a robust model validation for FPC (or FPT). The model validation for soil temperature seems acceptable and reasonable (Fig. 3), but, for the FPC, it seems very weak although the authors tried to simply compare the simulated PFT with a published national vegetation map of China (by Zheng et al, 2008). I am not clear how this has been done and how good is the comparison? I would suggest to use a better quantitative method like Kappa Statistic (Cohen, 1960) to improve the comparison; Cohen, Jacob (1960). A coefficient of agreement for nominal scales. Educational and Psychological Measurement 20 (1): 37–46.

**Response**: Thanks for this suggestion. Please refer to our response to the similar question by Reviewer #1.

2) lacking some discussions on the uncertainties of model key parameters (like "$CO_2$ fertilization effect" for 6 main PFT)? Because the $CO_2$ fertilization effect at PFT level seems a big challenge and remain large uncertainty in model simulations. It is not clear for me how the authors have considered and quantify the different response of 6 PFT to increasing $CO_2$ concentration over past 52 years in the model simulations? Did they use the same or different parameters?

**Response**: In terms of the $CO_2$ fertilization effect, Kimball (1983), Chang et al. (2016), Kim et al. (2016) and Schmid et al. (2016) stated that as $CO_2$ level increased, vegetation yield changed, and the change was however related to the environment conditions such as light, soil nutrient and soil moisture and temperature. In the manuscript we assume that the $CO_2$ fertilization effects can be reflected from the changes in photosynthesis and net primary productivity. In LPJ photosynthesis calculation follows the method proposed by Farquhar (1982) that was later modified by Collatz et al. (1991), Collatz et al. (1992) and Haxeltine and Prentice (1996). The aforementioned references are listed at the end of this paragraph. The parameters that are used for photosynthesis calculation and PFT-specifics are temperature inhibition function limiting photosynthesis at low (TLCO$_2$) and high

(TUCO$_2$) temperatures, leaf phenology such as growing degree days to attain full leaf cover (GDDs). The values of these parameters are presented in Table 1. After carbon assimilation, net primary productivity is calculated by subtracting the maintenance respiration from gross primary productivity where leaf C:N ratio, root C:N ratio and sap C:N ratio are used. These C:N ratios are kept constant for all PFTs though. Based on photosynthesis and net primary productivity calculations and the PFT specific parameters used in the calculation, it can be inferred that individual PFTs have different responses to CO$_2$ increase (see Table 5 below). As environmental conditions also affect the CO$_2$ fertilization effects, CO$_2$ increase does not necessarily result in the elevated net primary productivity as shown in Figure 8f and Table 5. Table 5 shows that different vegetation exhibits different responses to CO$_2$ increase, among them PAMD displays primarily positive response to CO$_2$ increase which also explains why PAMD has increased over a large portion of the study area during the past 52 years. Total FPC also shows positive response to the CO$_2$ increase, which is mostly dominated by the positive PAMD response. Admittedly, the LPJ simulation may not reflect the reality because the model keeps C:N ratios constant throughout the processes and the nitrogen effects on photosynthesis is simplified. This is certainly an area of further investigation and improvement.

Table 5. The response of individual PFT to CO$_2$ increase.

|  | Cells of positive response | Cells of negative response |
|---|---|---|
| FPC of TNEG | 367 | 140 |
| FPC of PAMD | 1567 | 100 |
| FPC of PASP | 671 | 658 |
| FPC of TSGS | 341 | 374 |
| Total FPC | 1596 | 152 |

Collatze G.J., Ball, J.T., Grivet C., Berry J.A.: Physiological and environmental-regulation of stomatal conductance, photosynthesis and transpiration – a model that includes a laminar boundary-layer, Agricultural and Forest Meteorology, 54, 107-136, 1991.

Collatze, G.J., Ribas-Carbo, M., and Berry J.A.: Coupled photosynthesis-stomatal conductance model for leaves of C4 plants, Australian Journal of Plant Physiology, 19, 519-538, 1992.

Chang, J., Ciais, P., Viovy, N., Vuichard, N., Herrero, M., Havlik, P., Wang, X., Sultan, B., and Soussana, J-F.: Effect of climate change, CO$_2$ trends, nitrogen addition, and land-cover and management intensity changes on the carbon balance of European grasslands, Global Change Biology, 22, 338-350, 2016.

Haxeltine A., Prentice, I.C.: A general model for the light-use efficiency of primary production, Functional Ecology, 10, 551-561, 1996.

Kim D., Oren, R., Qian, S. S.: Response to $CO_2$ enrichment of understory vegetation in the shade of forests, Global Change Biology, 22, 944-956, 2016.

Kimball, B.A: Carbon dioxide and agricultural yield: an assemblage and analysis of 430 prior observations, Agronomy Journal, 75,779-788, 1983.

Schmid, I., Franzaring, J., Muller, M., Brohon, N., Calvo, O.C., Hogy, P., and Fangmerier, H.: Effects of $CO_2$ enrichment and drought on photosynthesis, growth and yield of and old and a modern barley cultivar, Journal of Agronomy and Crop Science, 202, 81-95, 2016.

Specific points: 1) On page 13, line 276-279: I am not clear this mentioned comparison here. Please clarify this point by providing more information or explanations.
**Response:** Please see our response to your comment #1.

2) On page 14, line 311-317, Why you used the elasticity (E)? What is advantage of the E? Please also provide a key reference if possible.
**Response:** Elasticity is a non-parametric, robust and unbiased estimator (Sankarasubramanian et al. 2001; Elsner et al., 2010). Elasticity can be utilized to better measure the response of FPC to the changes in climate and soil conditions and so it is used in this study.

Sankarasubramanian, A., Vogel R.M., Limbrunner J.F.: Climate elasticity of streamflow in the United States, Water Resources Research, 37, 1771-1781, 2001

Elsner, M.M., Cuo, L., Voisin, N., Deems, J.S., Hamlet, A.F., Vano, J.A., Mickelson K.E.B., Lee S.-Y., Lettenmaier, D.P.: Implications of 21st century climate change for the hydrology of Washington State, 102, 225-260, 2010.

3) On page 17, line 382-385: It is too general statement here. How did you judge "it is largely consistent"? It would be better to use some quantitative indices or methods to justify this critical point here.
**Response**: Please refer to our response to Question #2 by Reviewer #1. We basically included more discussions with observation-based studies to clarify the statement.

4) For the Fig. 2 -3, There are missing some important information for the figure captions. I have no idea which line is model simulation and which is the observation? Please add this missing information directly in both figures.
**Response:** Sorry for this since for some reason the legends were removed after the figures were converted from the eps format to the pdf format. We will make sure that this problem won't occur in the revised version.

I would be happy to see this paper to be accepted after a major revision.